# Comprehensive Assessment of GPR68 Expression in Normal and Neoplastic Human Tissues Using a Novel Rabbit Monoclonal Antibody

**DOI:** 10.3390/ijms20215261

**Published:** 2019-10-23

**Authors:** Markus Herzig, Pooja Dasgupta, Daniel Kaemmerer, Jörg Sänger, Katja Evert, Stefan Schulz, Amelie Lupp

**Affiliations:** 1Institute of Pharmacology and Toxicology, Jena University Hospital, D-07747 Jena, Germany; markus.jakob.herzig@uni-jena.de (M.H.); Pooja.Dasgupta@med.uni-jena.de (P.D.); Stefan.Schulz@med.uni-jena.de (S.S.); 2Department of General and Visceral Surgery, Zentralklinik Bad Berka, D-99437 Bad Berka, Germany; daniel.kaemmerer@zentralklinik.de; 3Laboratory of Pathology and Cytology Bad Berka, D-99437 Bad Berka, Germany; pathologie.bad-berka@t-online.de; 4Department of Pathology, University of Regensburg, D-93053 Regensburg, Germany; Katja.Evert@ukr.de; 5Institute of Pathology, University Medicine of Greifswald, D-17475 Greifswald, Germany

**Keywords:** GPR68, OGR1, antibody, immunohistochemistry, neuroendocrine, tumours

## Abstract

GPR68 (OGR1) belongs to the proton-sensing G protein-coupled receptors that are involved in cellular adaptations to pH changes during tumour development. Although expression of GPR68 has been described in many tumour cell lines, little is known about its presence in human tumour entities. We characterised the novel rabbit monoclonal anti-human GPR68 antibody 16H23L16 using various cell lines and tissue specimens. The antibody was then applied to a large series of formalin-fixed, paraffin-embedded normal and neoplastic human tissue samples. Antibody specificity was demonstrated in a Western blot analysis of GPR68-expressing cells using specific siRNAs. Immunocytochemical experiments revealed pH-dependent changes in subcellular localisation of the receptor and internalisation after stimulation with lorazepam. In normal tissue, GPR68 was present in glucagon-producing islet cells, neuroendocrine cells of the intestinal tract, gastric glands, granulocytes, macrophages, muscle layers of arteries and arterioles, and capillaries. GPR68 was also expressed in neuroendocrine tumours, where it may be a positive prognostic factor, in pheochromocytomas, cervical adenocarcinomas, and endometrial cancer, as well as in paragangliomas, medullary thyroid carcinomas, gastrointestinal stromal tumours, and pancreatic adenocarcinomas. Often, tumour capillaries were also strongly GPR68-positive. The novel antibody 16H23L16 will be a valuable tool for basic research and for identifying GPR68-expressing tumours during histopathological examinations.

## 1. Introduction

GPR68, also known as ovarian cancer G protein-coupled receptor 1 (OGR1), is a member of the proton-sensing G protein-coupled receptor (GPCR) family that contains three additional members: GPR4, GPR65 (or T lymphocyte death-associated gene 8 protein, TDAG8), and GPR132 (or G2 accumulation protein, G2A). Unlike other receptors, the pH window of GPR68 is quite narrow. The receptor seems to be inactive at pH 7.8 and fully active at pH 6.8 [1,2], but the opposite pattern has also been shown [3]. The pH sensing ability of GPR68 is attributed to four histidines located in its extracellular loops. It is hypothesised that at a slightly alkaline pH, GPR68 is stabilised in an inactive state by hydrogen bonding of these residues. Protonation of the histidines at an acidic pH may cause loss of hydrogen bonding, thus allowing the receptor to adopt an active conformation [1]. In addition to direct activation by protons, ogerin and the benzodiazepine lorazepam have been shown to be positive allosteric modulators of GPR68 [4]. GPR68 is coupled to the G_q/11_-phospholipase-C/inositol-triphosphate (IP3) pathway, and upon activation, calcium is released from the endoplasmic reticulum into the cytosol [1].

Under many pathophysiological circumstances, such as inflammation, ischaemia, and tumour development, acidosis occurs in the local microenvironment, and proton-sensing GPCRs (amongst others) are involved in cellular adaptations to these changes [5,6,7]. In tumours, for instance, extracellular acidosis can cause increased tumour cell proliferation, migration, invasion, angiogenesis, and metastasis [6,7,8,9]. Accordingly, expression of proton-sensing GPCRs has been detected in many tumour cell lines [6,7]. However, data concerning the effects of GPR68 in these cells are contradictory. 

Studies have shown that GPR68 can act as either a tumour suppressor or a tumour promoter. In HEY human ovarian cancer cells, MCF-7 breast cancer cells, and Caco-2 colorectal adenocarcinoma cells, increased GPR68 expression is associated with reduced cell proliferation and migration [10,11,12], suggesting GPR68 may act as a tumour suppressor. GPR68 has also been described as a metastasis suppressor in studies involving orthotopic injection of human prostate PC-3 cells into athymic or NOD/SCID mice. However, in these experiments, no influence of GPR68 on primary tumour growth was observed [13]. In contrast, in the TRAMP (transgenic adenocarcinoma of the mouse prostate) cancer model, GPR68 deficiency led to reduced tumour formation compared to wild-type expression of GPR68 [14]. Additionally, after inoculation of melanoma B16-F10 cells into GPR68-deficient mice, decreased tumour size compared to wild-type animals was observed [15]. Thus, in these latter experiments, GPR68 acted as a tumour promoter. GPR68 expression has also been detected in human medulloblastoma tumour samples and in the DAOY medulloblastoma cell line and also in these cells a tumour promoting effect was observed [8]. Hence, the impact of GPR68 may vary depending on the tumour cell type, microenvironment, and there may be differences in the effects on primary tumors and metastases [6]. 

Unfortunately, other than its expression in tumour cell lines or in animal models, little is known about the presence of GPR68 in human tumour samples. This limited knowledge may be due to a lack of specific antibodies suited for use in formalin-fixed, paraffin-embedded tumour samples from routine pathological studies. Therefore, we have developed a novel rabbit monoclonal antibody, 16H23L16, which is directed against the carboxyl-terminal tail of human GPR68. 

In the present study, we demonstrate that this antibody is well suited both for Western blot analyses and immunocytochemistry in basic research and for immunohistochemical staining of routine clinical pathology samples. The antibody was further tested in a large series of formalin-fixed, paraffin-embedded normal and neoplastic human tissue samples, revealing GPR68 expression in glucagon-producing islet cells of the pancreas, specific endocrine cells of the intestinal tract, and pancreatic neuroendocrine tumours. Based on these findings, we expanded the number of neuroendocrine tumours studied and ultimately evaluated a broad panel of bronchopulmonary and gastroenteropancreatic neuroendocrine neoplasms of different origins for GPR68 expression. The staining results were then correlated with clinical data, such as grading, staging, glucagon or insulin secretion, and patient overall survival.

## 2. Results

### 2.1. Characterisation of the Rabbit Monoclonal Anti-Human GPR68 Antibody 16H23L16

The specificity of anti-human GPR68 antibody 16H23L16 was first tested by Western blot analysis. When membrane preparations from GPR68-transfected HEK-293 cells or BON-1 cells endogenously expressing GPR68 were electrophoretically separated and immunoblotted, the antibody recognized a broad band migrating at M_r_ 56,000–72,000 from both cell lines (Figure 1A, right panel, Figure 1B, right panel). This molecular weight corresponds well to the expected molecular weight of the glycosylated receptor [16]. In contrast, no immunosignal could be observed in mock-transfected HEK-293 cells (Figure 1A, left panel). GPR68 expression in BON-1 cells was then silenced by a specific siRNA to further confirm the specificity of the antibody. As expected, the immunosignal was distinctly decreased compared to that in with scrambled siRNA transfected cells (Figure 1B, left panel).

To corroborate literature reports of the pH sensitivity of GPR68, we additionally determined the subcellular localisation of the receptor at different pH levels by immunocytochemistry. When BON-1 cells were incubated in HEPES buffer at pH 6.8 or 7.4 for 60 min, complete internalisation of the receptor was observed (Figure 2A,B). In contrast, at pH 7.8, the immunosignal was primarily located at the plasma membrane (Figure 2C). After additional incubation of the cells for 30 min at pH 7.8 in the presence of 1 or 10 µM of the putative allosteric agonist lorazepam, complete internalisation of the receptor was again observed (Figure 2D). However, at lower concentrations (0.01 and 0.1 µM) of lorazepam, the receptor remained at the plasma membrane, confirming reported values for positive allosteric activity of lorazepam at GPR68 in the low micromolar range [4].

### 2.2. Immunohistochemical Detection of GPR68 Localisation in Normal Human Tissues

The rabbit monoclonal anti-GPR68 antibody 16H23L16 was used in immunohistochemical staining of various human non-neoplastic tissues (Figure 3 and Figure 4). In most cases, immunostaining was localised to the cytoplasm of the cells. A set of tissue samples showing positive staining for GPR68 was also incubated with 16H23L16 pre-adsorbed with its immunising peptide, which, in all cases, led to complete abolishment of the immunosignal (see insets in Figure 3A–C). Serial sections of GPR68-positive tissue samples were also stained with the polyclonal rabbit anti-OGR1 antibody ab61420 (Abcam), which is directed against the *N*-terminal tail of GPR68, and, thus, to another epitope of the receptor. These additional experiments revealed similar staining results as compared to those obtained with the novel monoclonal antibody 16H23L16. However, in comparison to the monoclonal antibody 16H12L16, the polyclonal antibody ab61420 detected less GPR68 receptors and caused a somewhat higher non-specific background staining (Figure 5).

As depicted in Figure 3A, very strong immunostaining of GPR68 was observed in distinct cell populations in the periphery, but not in the centre, of pancreatic islets. To more precisely differentiate GPR68-positive cells in the pancreatic islets, additional double-labelling experiments were performed for glucagon and insulin, which clearly revealed the presence of GPR68 in glucagon-secreting alpha cells, but not in insulin-producing beta cells (Figure 6). GPR68 was also expressed in neuroendocrine cells of the intestinal tract (Figure 3B), and in Brunner’s glands of the duodenum. Gastric glands of the antrum were also slightly GPR68-positive. In the ileum, epithelia at the tips of the villi showed strong GPR68 expression that was not observed in the other areas of the gastrointestinal tract (Figure 3C). A weak immunosignal was also present in the intramural ganglia of the gastrointestinal tract (Figure 3D). GPR68 expression was further detected in distinct cell populations present in the red pulp of the spleen and within the lumen of vessels of other organs, such as liver or stomach (Figure 3E,F). Based on their morphology, these cells probably represent granulocytes, and, as indicated by a co-expression with CD68 (see inset in Figure 3E), monocytes or macrophages. In contrast, the thymus and white pulp of the spleen were completely negative for GPR68 staining. GPR68 was also expressed in the smooth muscle layers of arteries and arterioles, and in capillaries; this was explicitly evident in lung tissue (Figure 4A–C), but was occasionally seen in other organs as well. Presence of GPR68 in the muscle layer of the arterial wall was verified by additional double-immunofluorescence analyses for alpha-smooth muscle actin and in capillaries by double-labelling experiments with CD31 (insets in Figure 4B,C). In lung tissue, single cells within the bronchial glands stained GPR68-positive (likely neuroendocrine cells), as did the muscle layers and epithelia of the larger airways. Strong GPR68 expression was observed in a subset of chondrocytes in larger airway cartilage and in bone tissue, the latter of which contained some GPR68-positive osteoblasts and osteocytes (Figure 4D,E). A strong GPR68 signal was additionally observed in mononuclear cells of the bone marrow (Figure 4F).

### 2.3. Immunohistochemical Detection of GPR68 Localisation in Different Human Tumour Entities

The patterns of GPR68 distribution in human tumour samples are summarised in Table 1. Representative examples of immunostaining are shown in Figure 7. Here, both cytoplasmic and membranous staining of cells was observed. Notably, GPR68 expression in the tumour samples displayed substantial inter- and intra-individual variability. Often, only small areas with strong staining were observed, whereas large regions of the tumours were completely devoid of any GPR68 expression, thus leading to a low overall rating (IRS values). Pronounced GPR68 expression associated with higher IRS values was seen especially in a subpopulation of neuroendocrine tumours, pheochromocytomas, adenocarcinomas of the cervix (in contrast to squamous cell carcinomas of the cervix, which were negative) and endometrial cancer. Weak expression was also observed in a high percentage of paragangliomas, medullary thyroid carcinomas, gastrointestinal stromal tumours, and pancreatic adenocarcinomas. Besides the tumour cells, especially in cases of lung cancer but also in some cases of other tumour entities such as prostate adenocarcinoma or lymphoma, the tumour capillaries were also strongly GPR68-positive. Particularly in gastric cancer, but occasionally in other tumour entities, GPR68-positive macrophages were present in tumour tissues. Furthermore, in one case of glioblastoma, accumulation of GPR68-positive granulocytes was observed.

As with normal tissue, a set of GPR68-positive tumour samples was additionally incubated with 16H23L16 pre-adsorbed with its immunising peptide. Again, a complete extinction of the immunosignal was observed (see insets in Figure 7B,D).

Regarding GPR68 expression in different tumour cell lines, a moderately strong immunosignal was obtained only with the neuroendocrine tumour cell line BON-1 and the ovarian carcinoma cell line A2780. In the hepatocellular carcinoma cell line HuH-7 and the breast cancer cell line BT474, weak to moderate GPR68 expression was noted. Only weak GPR68 immunosignals could be observed in the colon carcinoma cell line HT-29, the hepatocellular carcinoma cell line Hep-3B, the prostate cancer cell lines PC-3 and LNCaP, the breast cancer cell lines MCF-7 and MDA-MB-231, the cervical cancer cell line ME-180, and the epidermoid carcinoma cell line A431. All other cell lines investigated, the neuroblastoma cell lines SiMa, SK-N-SH and SH-SY5Y, the SCLC cell lines OH-1, NCI-H69 and NCI-H82, the lung adenocarcinoma cell line A549, the colon carcinoma cell lines SW480 and LoVo, the hepatocellular carcinoma cell lines HepG2 and SK-HEP-1, the renal cell carcinoma cell line A498, the urinary bladder carcinoma cell lines T24 and RT-112, the prostate cancer cell line DU145, the cervical cancer cell lines SW756, SiHa, CaSki and HeLa, and the ovarian carcinoma cell line SK-OV-3, were devoid of GPR68 expression.

### 2.4. GPR68 Expression in Bronchopulmonary and Gastroenteropancreatic Neuroendocrine Tumours

#### 2.4.1. GPR68 Expression Pattern

Figure 8 shows representative stainings of different bronchopulmonary and gastroenteropancreatic neuroendocrine tumours. Distinct immunostaining of the cytoplasm, but also of the plasma membrane of the tumour cells was observed. Also here, after preadsorption of 16H23L16 with its immunizing peptide a complete extinction of the immunosignal was noted (see insets in Figure 8A–C). Overall, regarding the percentage of signal-positive tumours (IRS ≥ 3) (Figure 9A) and the extent of expression (Figure 9B), GPR68 was most prominently expressed in neuroendocrine tumours from the pancreas, followed by those from the rectum or gut and typical carcinoids of the lung. However, expression levels again varied considerably between individual patients and sometimes between different samples from the same patient, which is illustrated by the lengths of the respective boxes and whiskers in Figure 9B. Here, e.g., in pancreatic tumours, IRS values ranged from 0 points (no expression) to 12 IRS points (maximum expression).

#### 2.4.2. Correlations with Clinical Data

When data from all bronchopulmonary and gastroenteropancreatic neuroendocrine tumours were analysed together, significant differences were noted between patients with and without lymph node metastases, with higher GPR68 IRS values in patients with no lymph node metastases (without lymph node metastases: 1.854 ± 0.294; with lymph node metastases: 1.398 ± 0.210; Mann–Whitney test: *p* = 0.039). Accordingly, Kaplan–Meier analysis revealed a slightly better outcome for patients with GPR68-positive tumours (IRS ≥ 3) compared to those with GPR68-negative neoplasms (log-rank test: *p* = 0.104; Figure 10A). Fittingly, a positive correlation was found between the IRS values of GPR68 and those of typical markers for neuroendocrine tumours, known to be associated with a good prognosis [17,18] (chromogranin A (rsp = 0.137, *p* = 0.028), somatostatin receptor (SST) 2A (rsp = 0.201, *p* = 0.001), SST3 (rsp = 0.133, *p* = 0.032), and SST5 (rsp = 0.148, *p* = 0.028)). Furthermore, significant differences regarding patient sex were observed, with lower GPR68 IRS values in males than in females (mean ± S.E.M: males: 1.226 ± 0.176, females: 1.856 ± 0.227; Mann–Whitney test: *p* = 0.017).

If only bronchopulmonary neuroendocrine tumours were considered, a positive association was detected between GPR68 expression and patient overall survival (rsp = 0.234, *p* = 0.035) and a negative correlation with levels of the proliferation marker Ki-67 (rsp = –0.222, *p* = 0.043). Also here, Kaplan–Meier analysis revealed a slightly better outcome for GPR68-positive tumour cases (log-rank test: *p* = 0.140; Figure 10B). Additionally, there was a positive correlation between the IRS levels of GPR68 and those of CgA (rsp = 0.294, *p* = 0.009), SST2 (rsp = 0.185, *p* = 0.094), and SST5 (rsp = 0.216, *p* = 0.050). Again, lower GPR68 IRS values were observed in males than in females (mean ± S.E.M: males: 0.731 ± 0.156, females: 1.615 ± 0.300; Mann–Whitney test: *p* = 0.023).

If only gastroenteropancreatic neuroendocrine tumours were included in the analysis, significantly higher GPR68 levels were again noted in patients without lymph node metastases (without lymph node metastases: 2.473 ± 0.512; with lymph node metastases: 1.449 ± 0.238; Mann–Whitney test: *p* = 0.012). Additionally, in gastroenteropancreatic neuroendocrine neoplasms, a tendency towards lower GPR68 levels was observed in the metastases as compared to the primary tumours (primary tumours: 1.833 ± 0.241, metastases: 1.286 ± 0.252; Mann–Whitney test: *p* = 0.089). However, Kaplan–Meier analysis could not demonstrate statistically significant differences between patients with GPR68-positive or -negative tumours, likely due to too few positive cases (log-rank test: 0.465; Figure 10C). Nevertheless, a positive association was shown between the presence of GPR68 and SST1 or SST2 expression (rsp = 0.211, *p* = 0.006; rsp = 0.191, *p* = 0.013, respectively). Corresponding to the findings in all tumours and in bronchopulmonary neoplasms alone, gastroenteropancreatic neuroendocrine tumours alone yielded slightly lower values in male patients than in female patients, though without reaching statistical significance (males: 1.486 ± 0.254, females: 2.025 ± 0.374; Mann–Whitney test: *p* = 0.127). 

If considering only the tumour entity with the highest percentage of GPR68-positive cases (pancreatic neuroendocrine neoplasms), a negative correlation was found between GPR68 expression and Ki-67 levels (rsp = -0.341, *p* = 0.020) or tumour grade (rsp = -0.269, *p* = 0.028), while a positive association between GPR68 and SST2 expression (rsp = 0.328, *p* = 0.024) was observed. Because double-labelling experiments revealed the presence of GPR68 in glucagon-producing, but not insulin-producing, pancreatic islet cells, pancreatic neuroendocrine tumours were additionally evaluated for insulin or glucagon expression. As expected, a strong positive relationship was found between IRS values for GPR68 and glucagon (rsp = 0.576, *p* < 0.001), but not insulin (rsp = 0.154, *p* = 0.330).

## 3. Discussion

### 3.1. Characterisation of the Rabbit Monoclonal Anti-Human GPR68 Antibody 16H23L16

Monoclonal antibodies are advantageous over polyclonal antibodies in that they are directed against a single epitope, which generally leads to more specific staining. Additionally, and most importantly, they are indefinitely available in an unlimited amount with consistent quality. Because no such antibody was commercially available, we aimed to develop a monoclonal anti-GPR68 antibody that could be used for Western blot analyses and immunocytochemistry in basic research, as well as for immunohistochemical staining of formalin-fixed, paraffin-embedded tissues in routine pathology. In the present study, we demonstrated that the carboxyl-terminal tail of human GPR68 can serve as an epitope for generating a rabbit monoclonal antibody that can be effectively used for all three applications, including immunofluorescence double-labelling of paraffin sections. 

We present strong evidence that the novel anti-GPR68 antibody 16H23L16 specifically detects its targeted receptor and does not cross-react with other proteins. First, in Western blot analyses, the anti-GPR68 antibody selectively detected its cognate receptor in crude extracts from GPR68-transfected HEK-293 cells and from BON-1 cells endogenously expressing the receptor, but not in extracts from mock-transfected HEK-293 cells. Second, immunoreactive band intensities were distinctly reduced after GPR68 knockdown with a specific siRNA. Third, the antibody revealed cytoplasmic staining of BON-1 cells at pH 6.8 and pH 7.4, with predominant cell-surface staining at pH 7.8, which corroborates previous data showing activation of the receptor at slightly acidic pH values and inactivation in slightly alkaline conditions [1,2]. Furthermore, we demonstrated internalisation of the receptor after incubation of the cells in the presence of the benzodiazepine lorazepam, which has been postulated to act as a positive allosteric modulator of GPR68 [4]. Fourth, the novel anti-GPR68 antibody 16H32L16 yielded highly efficient staining of distinct cell populations within formalin-fixed, paraffin-embedded human tissue samples, which was completely abolished after pre-adsorption of the antibody with its immunising peptide. Finally, when comparing the novel monoclonal antibody with a commercially available polyclonal antibody directed against the *N*-terminal tail of GPR68, similar staining results were obtained. With the novel monoclonal antibody, however, a more distinct immunosignal and less non-specific background staining were observed.

### 3.2. Immunohistochemical Detection of GPR68 Localisation in Normal Human Tissues

In the present investigation, strong staining for GPR68 of distinct cells within the pancreatic islets was observed. These were identified as glucagon-producing cells due to their anatomical localisation within the islets and through positive double-labelling of GPR68 and glucagon, but not insulin. Our results were further corroborated by a positive correlation between GPR68 and glucagon expression, but not insulin expression, in the pancreatic neuroendocrine tumours. These results contrast another study, in which expression of the receptor was postulated in beta cells based on the detection of GPR68 mRNA in the pancreatic beta cell line EndoC-βH2 [19]. However, our data are supported by another study that showed GPR68 mRNA expression in α-TC cells, a cell line derived from islet alpha cells [20]. The same study also reported lower glucagon production in GPR68-deficient mice compared to wild-type animals, but also decreased glucose-stimulated insulin secretion [20]. Because GPR68 is also present in neuroendocrine cells of the intestinal tract, this decreased glucose-stimulated insulin secretion may be explained by concomitant reduction of glucagon-like peptide-1 production in these mice.

In addition to neuroendocrine cells, GPR68 expression was also observed in gastric glands of the antrum, Brunner’s glands of the duodenum, and distinct parts of the ileal epithelium. At these sites, GPR68 may be involved in proton-dependent bicarbonate and mucin secretion.

GPR68 was also detected in distinct immune cell populations within the red pulp of the spleen and within blood vessels of different organs, most likely representing granulocytes and monocytes or macrophages by morphological appearance and by co-expression of the monocyte/macrophage marker CD68. The thymus and the white pulp of the spleen, in contrast, were GPR68-negative. These results support previous studies showing GPR68 mRNA expression in human spleens and in peripheral blood leukocytes and macrophages, but not in the thymus of either humans or FLAG-tagged GPR68-expressing mice [15,21]. Correspondingly, GPR68 has been detected in human neutrophils and in the neutrophil-like cell line HL-60 [22]. In myeloid cells, GPR68 expression has additionally been reported to be inducible by proinflammatory cytokines [23]. Moreover, GPR68-deficient mice have fewer peritoneal macrophages, as well as impaired macrophage function, compared to wild-type animals [15]. Furthermore, in an IL-10 knockout mouse model, GPR68 deficiency provides protection against the development of colitis [23]. 

In the current study, GPR68 expression was occasionally also observed in osteoblasts and osteocytes. Recent reports have indicated that GPR68 is involved in osteoclast differentiation, survival, and function, and it was speculated that increased osteoclast survival via GPR68 may contribute to bone loss during systemic and local acidosis [15,24]. Similarly, GPR68-deficient mice have fewer osteoclasts than wild-type animals, although they display no overt bone abnormalities [15]. GPR68 was also detected in the osteoblastic cell line NHOst, in which a GPR68-dependent increase in intracellular calcium and IP3 concentration, as well as enhanced COX-2 activity and PGE2 production upon acidification, was observed [25]. In the present study, GPR68 expression was also observed in bone marrow mononuclear cells. This finding corresponds well with previous data on GPR68 expression in bone marrow [15], specifically, in bone marrow mononuclear cells [26], and in the pre-osteoclast-like cell line RAW 264.7 [26]. 

We also detected GPR68 expression in a subpopulation of chondrocytes of both bone and airway cartilages. Similarly, in literature, GPR68 has been demonstrated to be present in rat lumbar endplate chondrocytes and it is probably involved in acid-induced apoptosis of these cells [27].

In the present study, distinct staining of GPR68 was also found in smooth muscle layers of arteries and arterioles and in capillaries of various organs, particularly in lung tissue, corroborating other data obtained from mice with FLAG-tagged GPR68 [15] or from in situ hybridisation experiments [28]. The exceptionally high percentage of GPR68-positive vessels and GPR68 expression in the epithelium and musculature of airways may also explain the GPR68 expression detected at the mRNA level in lung tissue [21,29]. Expression of GPR68 in airway smooth muscle cells was demonstrated previously [30,31,32], and it was shown that (via GPR68) extracellular acidification stimulates IL-6 synthesis [30], connective tissue growth factor production [31], and contraction [32] of these cells. Additionally, acid-induced mucin 5AC hypersecretion by lung epithelial cells is seemingly triggered via GPR68 [33]. Thus, GPR68 may be involved in airway inflammation and remodelling, e.g., during asthma. Fittingly, GPR68-deficient mice lack the cardinal features of asthma, including airway eosinophilia and hyperresponsiveness as well as goblet cell metaplasia, in association with substantial inhibition of Th2-related cytokines and IgE production in an ovalbumin-induced asthma model [34].

### 3.3. Immunohistochemical Detection of GPR68 Localisation in Human Neoplastic Tissues

Few reports of GPR68 expression in human tumours have been published to date. To our knowledge, only human pancreatic tumours [35], medulloblastomas [8], and certain dermal tumours, such as Merkel-cell carcinomas, dermatofibrosarcomas protuberans, atypical fibroxanthomas, and pleomorphic dermal sarcomas [36], have been investigated for possible GPR68 expression so far. Thus, the present study comprehensively examined GPR68 expression in a broad range of human tumour entities. Interestingly, and likely because GPR68 is also expressed in non-neoplastic endocrine and neuroendocrine cells, a high percentage of GPR68-positive cases was observed in tumours of endocrine or neuroendocrine origin, such as growth hormone-producing pituitary adenomas, bronchopulmonary and gastroenteropancreatic neuroendocrine tumours, pheochromocytomas, paragangliomas, and medullary thyroid carcinomas. Therefore, it was expected that the most robust GPR68 expression in all cell lines tested was observed in the neuroendocrine tumour cell line BON-1. Nevertheless, we also confirmed previous reports of weak GPR68 expression in the breast cancer cell lines MDA-MB-231 (30) and MCF-7 [11] and in the prostate cancer cell line PC-3 [13]. However, the overall intensity of GPR68 expression in the human tumour entities and in the cell lines was relatively low. Thus, GPR68 is clearly of no relevance as a diagnostic or therapeutic target in all these tumour entities, including bronchopulmonary and gastroenteropancreatic neuroendocrine tumours. Only in a few cases of pancreatic (likely glucagon-producing) neuroendocrine tumours with exceptionally high expression GPR68 may be of some interest in this respect.

Among the large panel of neuroendocrine tumours investigated, differences in GPR68 expression were observed with respect to the localisation and malignancy of the tumour entities, with higher IRS values in tumours of the pancreas, rectum, or gut compared to those from other sites. Additionally, higher GPR68 expression was noted in well-differentiated tumours than in highly aggressive neoplasms with high Ki-67 values and with lymph node metastases already present at diagnosis. Moreover, a positive association between GPR68 expression and favourable patient outcomes was observed. Interestingly, sex-associated differences were observed, with higher overall GPR68 expression, but also a higher overall survival rate in female than in male patients. Together, our findings indicate that in neuroendocrine tumours, GPR68 may act as a tumour suppressor. Further investigations in other (neuro)endocrine tumour entities with a high prevalence of GPR68, such as pheochromocytomas, paragangliomas, and medullary thyroid carcinomas, but also with cervical adenocarcinomas and endometrial cancer will be of interest in this respect.

Independent of tumour cells, capillaries, granulocytes, and macrophages within some tumours were strongly GPR68-positive. Similar findings have been described in the literature [14,16,37], and the contribution of GPR68 expression in tumour stroma on cancer growth has been demonstrated in pancreatic cancer [35]. Therefore, indirect targeting of these tumours via GPR68 expressed in the tumour stroma may also represent a promising therapeutic strategy [35].

## 4. Materials and Methods

### 4.1. Antibody

A rabbit monoclonal antibody (16H23L16) that recognizes the carboxyl-terminal tail of human GPR68 was generated in collaboration with and obtained from Thermo Fisher Scientific (Waltham, MA, USA). The identity of the peptide used for immunisations of the rabbits was Cys-TKLHPAFQTPNSPGSGGFPTGRLA, corresponding to residues 342–365 of human GPR68. This sequence is unique for human GPR68 and, because of dissimilarities in the carboxyl-terminal amino acid sequence, 16H23L16 does not cross-react with rat or mouse GPR68 (Figure 11).

### 4.2. Western Blot Analysis

Human embryonic kidney 293 cells (HEK-293; DMSZ, Braunschweig, Germany), either mock-transfected or transfected with human GPR68, or BON-1 cells (DMSZ, Braunschweig, Germany) endogenously expressing GPR68 were seeded onto poly-L-lysine-coated 60-mm dishes and grown to 80% confluency. Cells were lysed in detergent buffer (20 mM HEPES (pH 7.4), 150 mM NaCl, 5 mM EDTA, 1% Triton X-100, 10% glycerol, 0.1% SDS, 0.2 mM phenylmethylsulfonylfluoride, 10 mg/mL leupeptin, 1 mg/mL pepstatin A, 1 mg/mL aprotinin, and 10 mg/mL bacitracin). GPR68 was enriched using wheat germ lectin agarose beads. Samples were subjected to 7.5% SDS-polyacrylamide gel electrophoresis and immunoblotted onto polyvinylidene fluoride (PVDF) membranes. Blots were incubated with the rabbit monoclonal anti-GPR68 antibody 16H23L16 at a dilution of 1:1000 followed by 1:5000-diluted peroxidase-conjugated secondary anti-rabbit antibody (Santa Cruz Biotechnology, Dallas, TX, USA) and enhanced chemiluminescence detection (Amersham, Braunschweig, Germany).

When indicated, expression of endogenous GPR68 in BON-1 cells was silenced using chemically synthesised double-stranded GPR68 siRNA duplexes (Santa Cruz Biotechnology, Dallas, TX, USA) according to the manufacturer’s instructions. As negative control, scrambled siRNA was used (Santa Cruz Biotechnology, Dallas, TX, USA).

### 4.3. Immunocytochemistry

BON-1 cells were grown on coverslips overnight and then exposed to different pH conditions (pH 6.8, 7.4, or 7.8) for 60 min. When indicated, cells were additionally stimulated with different concentrations of lorazepam (0.01, 0.1, 1, or 10 µM) for 30 min at pH 7.8. Cells were fixed with 4% paraformaldehyde and 0.2% picric acid in phosphate buffer (pH 6.9) for 20 min at room temperature, washed with phosphate buffer, and incubated with 16H23L16 (1:1000) at room temperature for 2 h, followed by an Alexa Fluor 488-conjugated secondary antibody (Invitrogen, Karlsruhe, Germany; dilution:1:5000) overnight at 4 °C. Samples were mounted and examined using a Zeiss LSM 510 META laser scanning confocal microscope (Jena, Germany). 

### 4.4. Immunohistochemical Evaluation of GPR68 Expression in Normal and Neoplastic Human Tissues and Tumour Cell Lines

#### 4.4.1. Tumour Specimens

For the initial evaluation of GPR68 expression in different human tumour entities, 265 formalin-fixed and paraffin-embedded tumour samples (neuroendocrine tumours, originally 10 samples; Table 1) were obtained from the Department of Pathology of the Ernst-Moritz-Arndt-University (Greifswald, Germany) and the Laboratory of Pathology and Cytology Bad Berka (Bad Berka, Germany). Many of the tumour specimens contained adjacent non-malignant tissue that was analysed as well. Additionally, tumour-free human tissue samples from cortex, lung, heart, liver, gallbladder, stomach, gut, pancreas, kidney, spleen, tonsils, thymus, lymph nodes, testicles, placenta, and bone (*n* = 6–15 each), obtained from the Department of Pathology of the Ernst-Moritz-Arndt-University (Greifswald, Germany) and the Laboratory of Pathology and Cytology Bad Berka (Bad Berka, Germany), were also evaluated. Staining patterns were compared to those seen in the non-malignant tissues surrounding the tumours.

#### 4.4.2. Bronchopulmonary and Gastroenteropancreatic Neuroendocrine Neoplasms

##### Tumour Samples

For the subsequent assessment of GPR68 expression in bronchopulmonary and gastroenteropancreatic neuroendocrine neoplasms, 713 tumour samples from 278 patients (in detail, 123 × 1, 52 × 2, 42 × 3, 25 × 4, 13 × 5, 8 × 6, 3 × 7, 6 × 8, 1 × 9, 2 × 12, 2 × 14, and 1 × 17 samples per patient; 374 primary tumour samples from 198 patients, 277 metastasis samples from 107 patients, and 62 unknown samples from 45 patients) were included in the present investigation. From some patients, both primary and metastatic samples were available. Of the tumours, 91 (32.7%) originated from the lung (in detail: 24 typical carcinoids (TC), 27 atypical carcinoids (AC), 39 small-cell lung cancer (SCLC), and 1 large-cell neuroendocrine carcinoma), 18 (6.5%) from the stomach, 15 (5.4%) from the duodenum/jejunum, 58 (20.9%) from the ileum, 5 (1.8%) from the appendix, 9 (3.2%) from the colon, 15 (5.4%) from the rectum, and 47 (16.9%) from the pancreas. Localisation of 20 (7.2%) primary tumours was unknown. The samples were provided by the Institute of Pathology and Cytology Bad Berka (Bad Berka, Germany), and had been surgically removed between 1998 and 2016 at the Department of General and Visceral Surgery, Zentralklinik Bad Berka (Bad Berka, Germany). Clinical data were gathered from patient records.

Permission was obtained from the local ethics committee (Ethikkommission der Landesärztekammer Thüringen; approval number: ha/1005/09/112; 04.02.2009) for this retrospective analysis. All data were recorded and analysed anonymously.

##### Patient Characteristics

The neuroendocrine tumours evaluated in the present investigation were from 140 male (50% of the cases) and 121 female (44%) patients. The sex of 17 patients (6%) was unknown. The overall mean age of the patients at diagnosis was 58.8 years (median: 60.0 years, range: 12.1–83.9 years). Forty-eight of the (corresponding) primary tumours (17%) were classified as T1; 41 (15%) as T2; 58 (21%) as T3, and 24 (9%) as T4. In 107 cases (38%), the extent of the primary tumour was unknown. Lymph node metastases were confirmed in 120 cases (44%), while seventy-nine patients (28%) showed none. For 79 tumours (28%), the patient’s lymph node status was not known. Eighty-one patients (29%) had no distant metastases, 113 patients (41%) had distant metastases, and 84 patients (30%) lacked any report of distant metastases in their medical files. At diagnosis, 17 of the 167 patients with gastroenteropancreatic neuroendocrine neoplasm (GEP-NEN) (10%) had UICC stage I disease, 12 patients (7%) had stage II disease, 22 patients (13%) had stage III disease, and 99 patients (60%) had stage IV disease. For 17 GEP-NEN patients (10%), the disease stage was unknown. For the 91 bronchopulmonary tumour patients and the 20 patients with unknown tumour origin, the stage of the disease was not determined. Regarding histological grading, 102 patients (37%) had grade 1, 86 patients (31%) had grade 2, and 63 patients (23%) displayed grade 3 histology. Tumour grading was unknown for 27 (9%) patients. 

The median follow-up time was 52.5 months overall. One hundred sixty-three patients were alive at the end of the follow-up period, 35 patients had no survival data available, and 80 patients died of tumour-related causes; of these, the median survival time was 26.5 months, which differed according to patient sex (19.5 months for males vs. 39.3 months for females; Mann–Whitney test: *p* = 0.016). 

#### 4.4.3. Cell Lines and Cytoblocks

Neuroblastoma cell lines (SiMa, SK-N-SH, and SH-SY5Y), SCLC cell lines (OH-1, NCI-H69, and NCI-H82), a lung adenocarcinoma cell line (A549), colon carcinoma cell lines (SW480, HT-29, and LoVo), hepatocellular carcinoma cell lines (HepG2, Hep3B, SK-HEP-1, and HuH-7), a renal cell carcinoma cell line (A498), a neuroendocrine tumour cell line (BON-1), urinary bladder carcinoma cell lines (T24 and RT-112), prostate cancer cell lines (PC-3, DU145, and LNCaP), breast cancer cell lines (MDA-MB-231, MCF-7, and BT-474), cervical cancer cell lines (SW756, SiHa, ME-180, CaSki, and HeLa), ovarian carcinoma cell lines (SK-OV-3 and A2780), and an epidermoid carcinoma cell line (A431) (DMSZ, Braunschweig, Germany) were grown in 75-cm² culture flasks to 80% confluency. For preparation of cytoblocks, cells were washed once with phosphate-buffered saline and transferred into 10% buffered formalin for 2 h. After centrifugation for 10 min at 3500× *g*, the supernatant was removed, and 1 mL of pooled human plasma was added to the cell samples. After brief vortexing, 100 µl fibrinogen was added to each sample and samples were vortexed again. The resulting clots were incubated for another 24 h in 10% buffered formalin and then embedded in paraffin blocks.

#### 4.4.4. Immunohistochemistry

From the paraffin blocks, 4-µm sections were prepared and floated onto positively-charged slides. Immunostaining was performed by an indirect peroxidase labelling method as described previously [38]. Briefly, sections were dewaxed, microwaved in 10 mM citric acid (pH 6.0) for 16 min at 600 W, and incubated with 16H23L16 (dilution: 1:500) overnight at 4 °C. For comparison, a subset of serial sections was additionally incubated with the rabbit polyclonal anti-OGR1-antibody ab61420, directed against the *N*-terminal tail of GPR68 (dilution: 1:50; Abcam, Cambridge, UK). Detection of the primary antibodies was performed using biotinylated anti-rabbit IgG followed by incubation with peroxidase-conjugated avidin (Vector ABC “Elite” kit; Vector Laboratories, Burlingame, CA, USA). Binding of the primary antibody was visualised using 3-amino-9-ethylcarbazole in acetate buffer (BioGenex, San Ramon, CA, USA). Sections were rinsed, counterstained with Mayer’s haematoxylin, and mounted in Vectamount™ mounting medium (Vector Laboratories, Burlingame, CA, USA). For immunohistochemical controls, 16H23L16 was either omitted or adsorbed for 2 h at room temperature with 10 µg/mL of the peptide used for immunisations.

For dual immunofluorescence, sections were incubated overnight at 4 °C with 16H23L16 together with either a mouse monoclonal anti-insulin antibody (1:100; Abcam, Cambridge, MA, USA), a mouse monoclonal anti-glucagon antibody (1:500; Sigma-Aldrich, St. Louis, MO, USA), a mouse monoclonal anti-CD68 antibody (1:500; Abcam, Cambridge, UK), a mouse monoclonal anti-smooth muscle actin antibody (1:400; Sigma-Aldrich, St. Louis, MO, USA) or a mouse monoclonal anti-CD31 antibody (1:1000; DAKO, Glostrup, Denmark). Sections were incubated with Cy3-conjugated anti-rabbit and Alexa Fluor 488-conjugated anti-mouse antibodies. Specimens were mounted and examined using a Zeiss LSM 510 META laser scanning confocal microscope.

Staining of GPR68 in the tumour slides was scored with the semiquantitative Immunoreactivity Score (IRS) according to Remmele and Stegner [39]. The percentage of positive tumour cells quantified in five gradations (no positive cells (0), <10% positive cells (1), 10–50% positive cells (2), 51–80% positive cells (3), and >80% positive cells (4) was multiplied by the staining intensity quantified in four gradations (no staining (0), mild staining (1), moderate staining (2), and strong staining (3)). Thus, IRS values ranging from 0 to 12 were obtained. For neuroendocrine tumours, several patients had more than one tumour slide, so the arithmetic mean was calculated from the IRS values of the different slides belonging to the same patient. All immunohistochemical stainings were evaluated by two independent blinded investigators (MH, AL). For discrepant scores, final decisions were achieved by consensus.

#### 4.4.5. Statistics

For statistical analysis, SPSS 25.0.0.0 (IBM, Armonk, NY, USA) was used. Because the data were not normally distributed (Kolmogorov–Smirnov test), Kruskal–Wallis test, Mann–Whitney test, Chi-square test, Kendall’s τ-b test, and Spearman’s rank correlation were performed. For survival analysis, the Kaplan–Meier method with a log-rank test was used. *p* values ≤0.05 were considered statistically significant.

## 5. Conclusions

We have generated and characterised a novel rabbit monoclonal anti-human GPR68 antibody that is well suited for Western blot and immunocytochemical analyses in basic research and for visualising human GPR68 in formalin-fixed, paraffin-embedded tissues during pathological examinations.

This antibody enabled detection of GPR68 expression in a wide variety of non-neoplastic and neoplastic tissues as well as cell populations. Our results also point to a possible involvement of GPR68 in many important physiological and pathophysiological processes, such as hormone secretion, glucose homeostasis, regulation of bicarbonate and mucus secretion in the gastrointestinal tract, modulation of inflammatory processes, angiogenesis, and tumour growth. However, further experiments are clearly necessary to substantiate these findings.

## Figures and Tables

**Figure 1 ijms-20-05261-f001:**
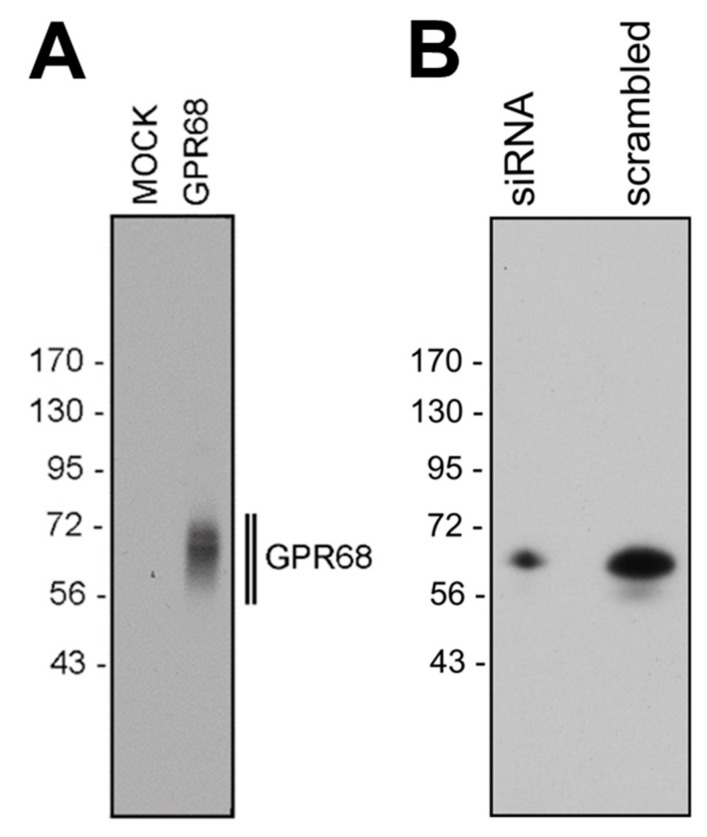
Specificity analysis of the monoclonal rabbit anti-human GPR68 antibody 16H23L16. (**A**) Western blot analysis of whole-cell preparations from mock- or stably GPR68-transfected HEK-293 cells. (**B**) Western blot analysis of BON-1 cells endogenously expressing GPR68 after transfection with scrambled siRNA (scrambled, right lane) or with specific GPR68 siRNA (left lane), causing downregulation of GPR68 expression. Ordinate, migration of protein molecular weight markers (kDa). Representative results from one of three independent experiments are shown.

**Figure 2 ijms-20-05261-f002:**
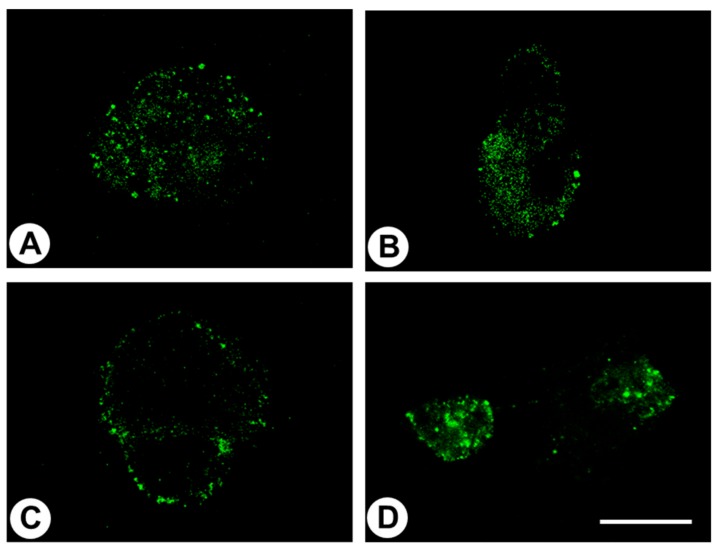
Immunocytochemical analysis of BON-1 cells endogenously expressing GPR68. Cells were exposed for 60 min to different pH conditions and were subsequently fixed and immunofluorescently stained with the anti-GPR68 antibody 16H23L16. Note that at pH 6.8 or pH 7.4, GPR68 was present in the cytosol (**A**,**B**), whereas at pH 7.8, the immunofluorescence was localised predominantly to the plasma membrane (**C**). After stimulation of the cells at pH 7.8 with 10 µM lorazepam, the immunosignal was again localised to the cytosol (**D**). Representative results from one of three independent experiments are shown. Scale bar: 20 µm.

**Figure 3 ijms-20-05261-f003:**
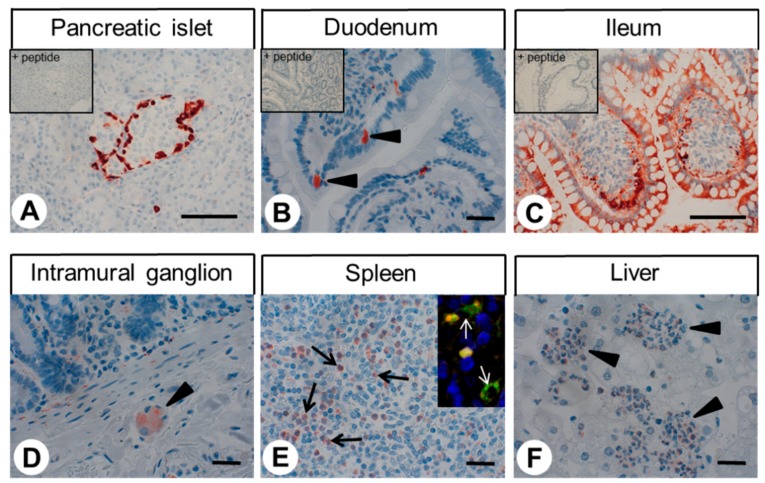
Immunohistochemical detection of GPR68 localisation in different normal human tissues (I). Immunohistochemical staining (red-brown colour), counterstaining with haematoxylin. Scale bar: 100 µm (**A**,**C**) and 30 µm (**B**,**D**–**F**). Insets in (**A**–**C**): For adsorption controls, the anti-GPR68 antibody 16H23L16 was incubated with 10 µg/mL of the peptide used for immunisations. Arrowheads in (**B**): positive neuroendocrine cells; arrowhead in (**D**): positive intramural ganglion in the duodenum; arrows in (**E**): positive immune cells in the spleen; arrowheads in (**F**): positive immune cells in the liver, from the shape of their nuclei probably representing granulocytes. Inset in (**E**): double-labelling immunohistochemical analysis of human spleen tissue. Samples were incubated with the rabbit monoclonal anti-GPR68 antibody 16H23L16 together with a mouse monoclonal anti-CD68 antibody. Labelling for GPR68 was visualised using a Cy3-conjugated anti-rabbit antibody (red), and immunostaining for CD68 was envisioned using an Alexa Fluor 488-conjugated anti-mouse antibody (green). White arrows indicate the double-labelled cells.

**Figure 4 ijms-20-05261-f004:**
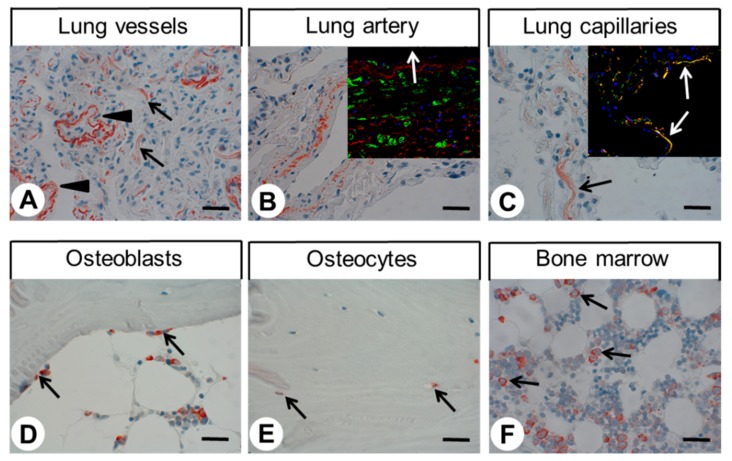
Immunohistochemical detection of GPR68 localisation in different normal human tissues (II). Immunohistochemical staining (red-brown colour), counterstaining with haematoxylin. Scale bar: 30 µm (**A**–**F**). Arrowheads in (**A**): positive muscle layers of arterioles, arrows in (**A**): positive capillaries; arrow in (**C**): positive capillary; arrows in (**D**) positive osteoblasts; arrows in (**E**): positive osteocytes; arrows in (**F**): positive mononuclear cells. Insets in (**B,C**): double-labelling immunohistochemical analysis of human lung tissue. Samples were incubated with the rabbit monoclonal anti-GPR68 antibody 16H23L16 together with either a mouse monoclonal anti-smooth muscle actin antibody (**B**) or a mouse monoclonal anti-CD31 antibody (**C**). Labelling for GPR68 was visualised using a Cy3-conjugated anti-rabbit antibody (red), and immunostaining for alpha smooth muscle actin or CD31 was envisioned using an Alexa Fluor 488-conjugated anti-mouse antibody (green). White arrow in (**B**) indicates the lumen of the artery; white arrows in (**C**) indicate double-labelled capillaries.

**Figure 5 ijms-20-05261-f005:**
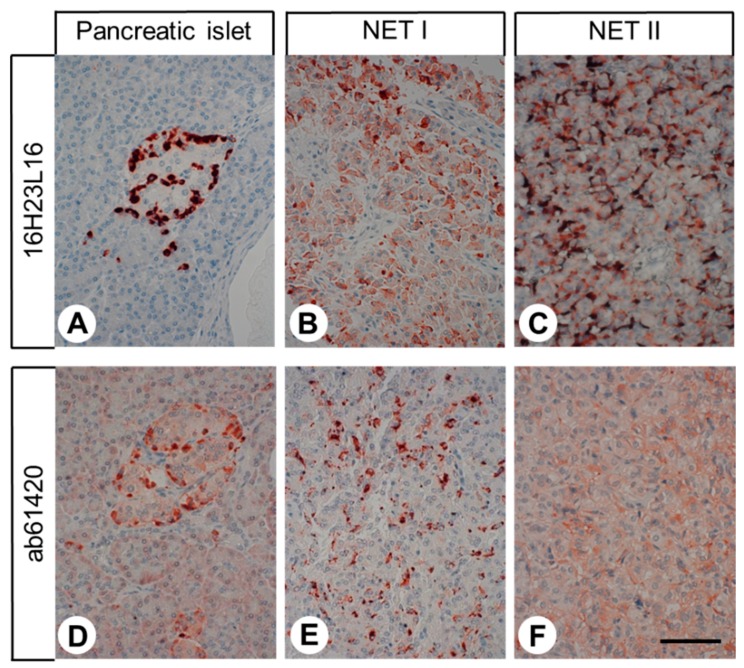
Comparative immunohistochemical stainings with the novel monoclonal rabbit anti-GPR68 antibody 16H23L16 or with the polyclonal rabbit anti-OGR1 antibody ab61420 (Abcam). Immunohistochemical staining (red-brown colour), counterstaining with haematoxylin. Scale bar: 100 µm (**A**–**F**). NET: neuroendocrine tumour.

**Figure 6 ijms-20-05261-f006:**
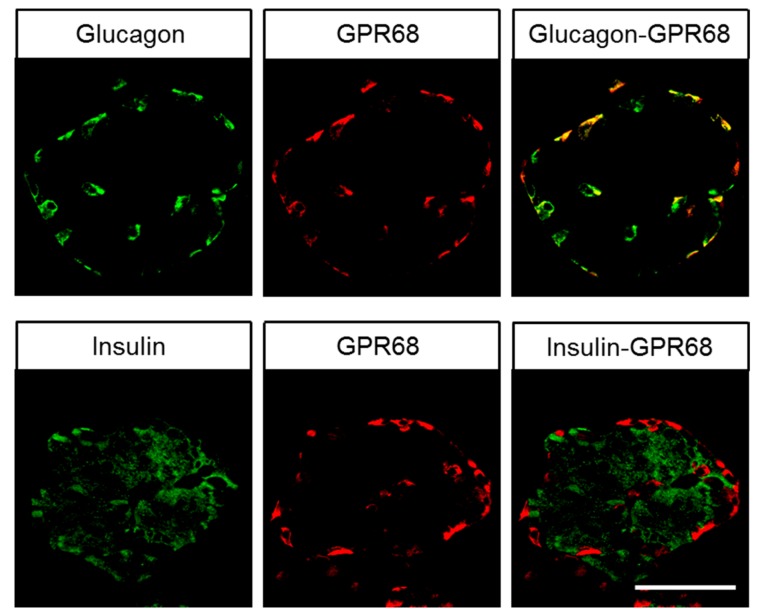
Double-labelling immunohistochemical analysis of human pancreatic islets. Sections were dewaxed and microwaved in citric acid. Adjacent sections of each tissue sample were incubated with the rabbit monoclonal anti-GPR68 antibody 16H23L16 together with mouse monoclonal antibodies against insulin or glucagon. Labelling for GPR68 was visualised using a Cy3-conjugated anti-rabbit antibody (red), and labelling for insulin and glucagon was envisioned using an Alexa Fluor 488-conjugated anti-mouse antibody (green). Scale bar: 100 µm.

**Figure 7 ijms-20-05261-f007:**
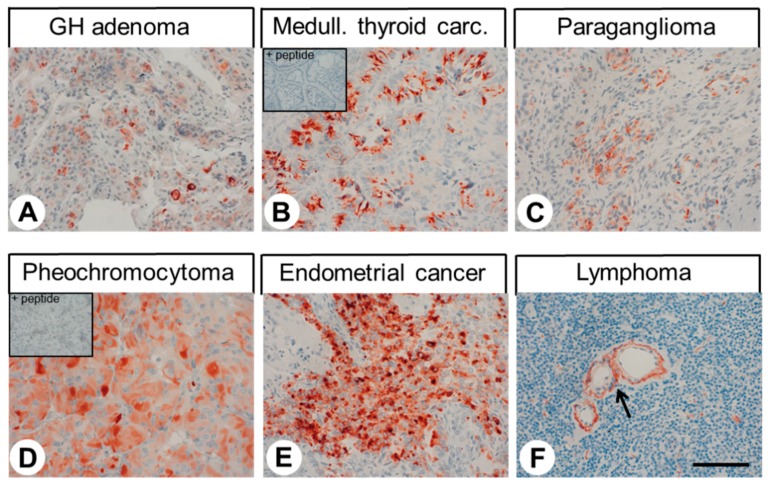
Immunohistochemical detection of GPR68 localisation in different human tumour entities. Immunohistochemical staining (red-brown colour), counterstaining with haematoxylin. Scale bar: 100 µm (**A**–**F**). Insets in (**B**) and (**D**): for adsorption controls, the anti-GPR68 antibody 16H23L16 was incubated with 10 µg/mL of the peptide used for immunisations. Arrow in (**F**): GPR68-positive vessels.

**Figure 8 ijms-20-05261-f008:**
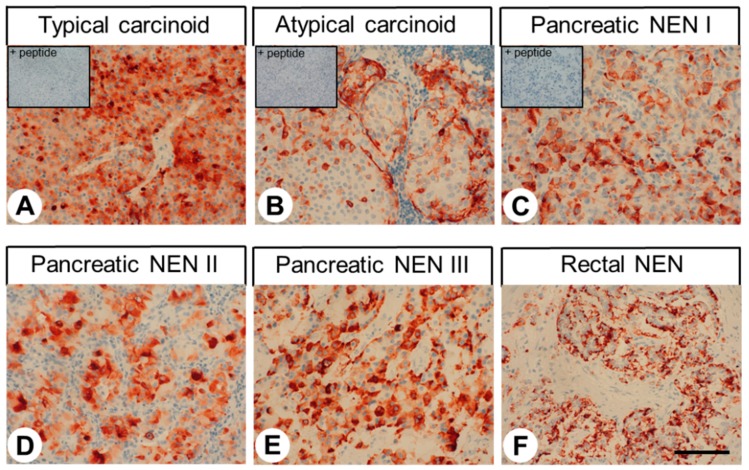
GPR68 expression pattern in different bronchopulmonary and gastroenteropancreatic neuroendocrine tumour entities. Immunohistochemical staining (red-brown colour), counterstaining with haematoxylin. Scale bar: 100 µm (**A**–**F**). Insets in (**A**–**C**): for adsorption controls, the anti-GPR68 antibody 16H23L16 was incubated with 10 µg/mL of the peptide used for immunisations. NEN: neuroendocrine neoplasm.

**Figure 9 ijms-20-05261-f009:**
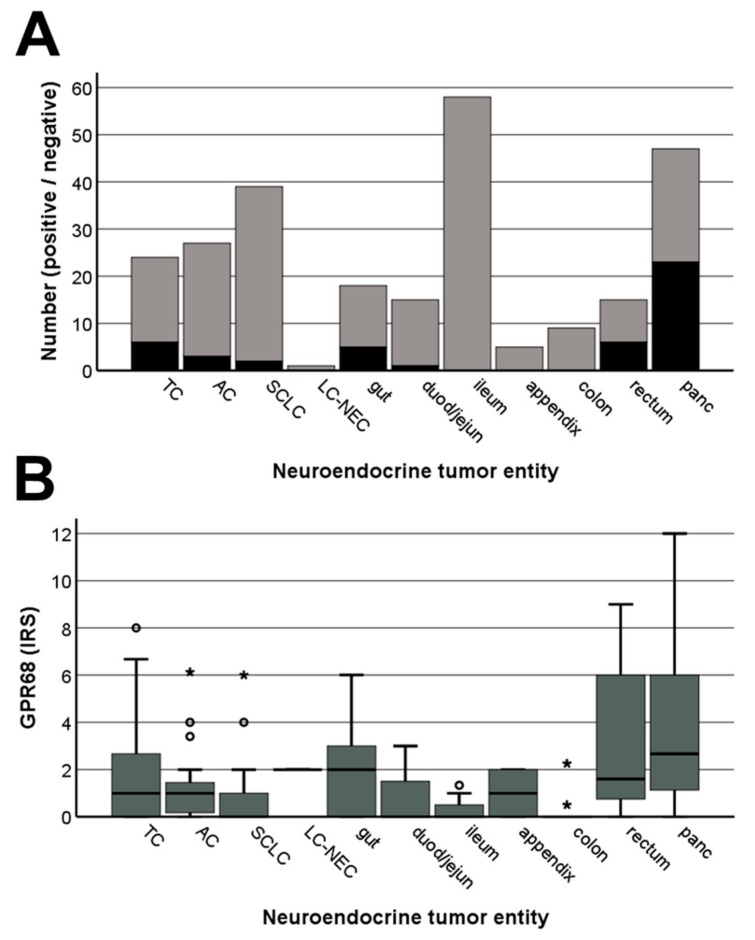
Expression profile of GPR68 in different bronchopulmonary and gastroenteropancreatic neuroendocrine tumor entities. (**A**) Number of GPR68-positive (black) and GPR68-negative cases (grey) within the different neuroendocrine tumour entities. Tumours were only considered positive at Immunoreactivity Score (IRS) values ≥3. (**B**) GPR68 expression levels (IRS values) in the different neuroendocrine tumour entities. Depicted are median values, upper and lower quartiles, minimum and maximum values, and outliers. Outliers were defined as follows: circles, mild outliers (data points between 1.5- and 3-times above the upper quartile or below the lower quartile); asterisks, extreme outliers (data that fell more than 3 times above the upper quartile or below the lower quartile). TC, typical carcinoid of the lung; AC, atypical carcinoid of the lung; SCLC, small-cell lung cancer; LC-NEC: large-cell neuroendocrine carcinoma; gut, gastroenteropancreatic neuroendocrine tumor (GEP-NEN) from the gut; duod/jejun, GEP-NEN from the duodenum or jejunum; ileum, GEP-NEN from the ileum; colon, GEP-NEN from the colon; rectum, GEP-NEN from the rectum; panc, pancreatic neuroendocrine tumour.

**Figure 10 ijms-20-05261-f010:**
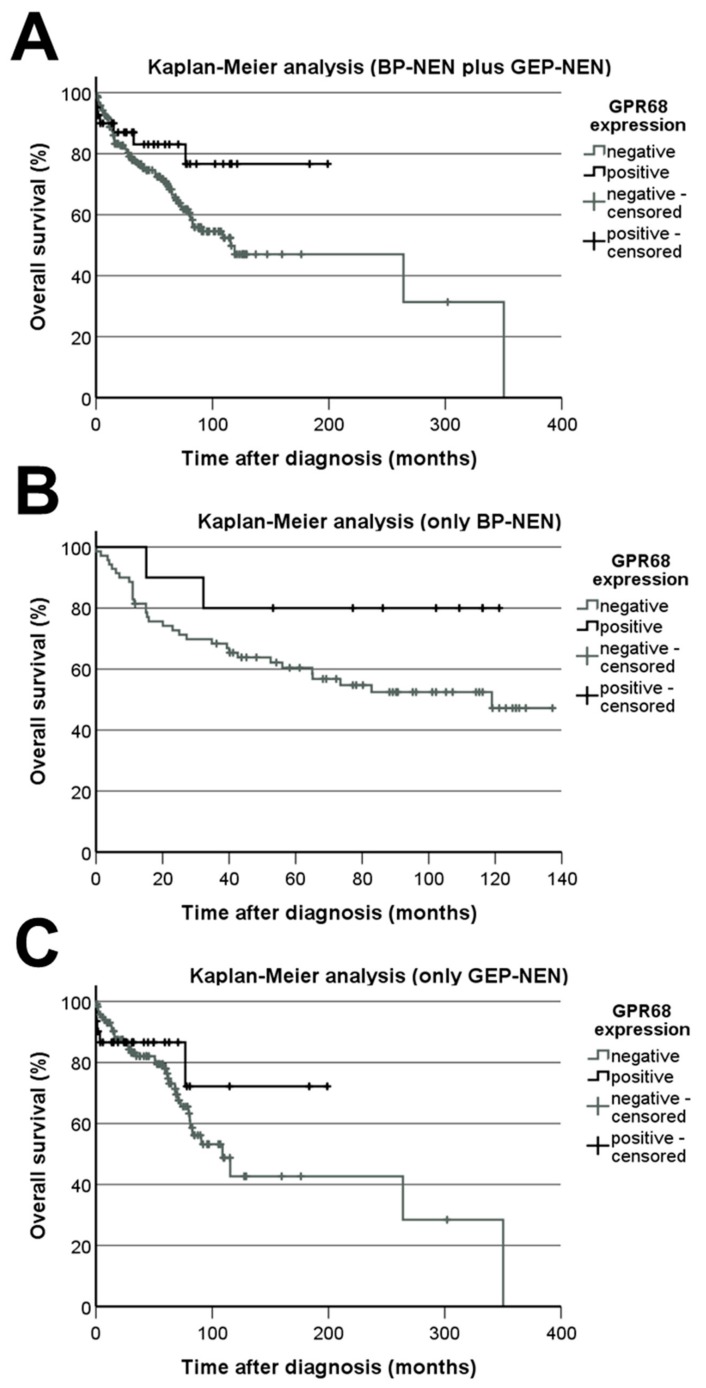
GPR68 expression-related overall survival of patients. Kaplan–Meier analysis of patient survival with respect to GPR68-positive and -negative tumours. (**A**) Bronchopulmonary tumours (BP-NEN) plus gastroenteropancreatic neuroendocrine tumours (GEP-NEN). (**B**) Only BP-NEN. (C) Only GEP-NEN. Log-rank test: *p* = 0.104 (**A**), *p* = 0.140 (**B**), and *p* = 0.465 (**C**).

**Figure 11 ijms-20-05261-f011:**
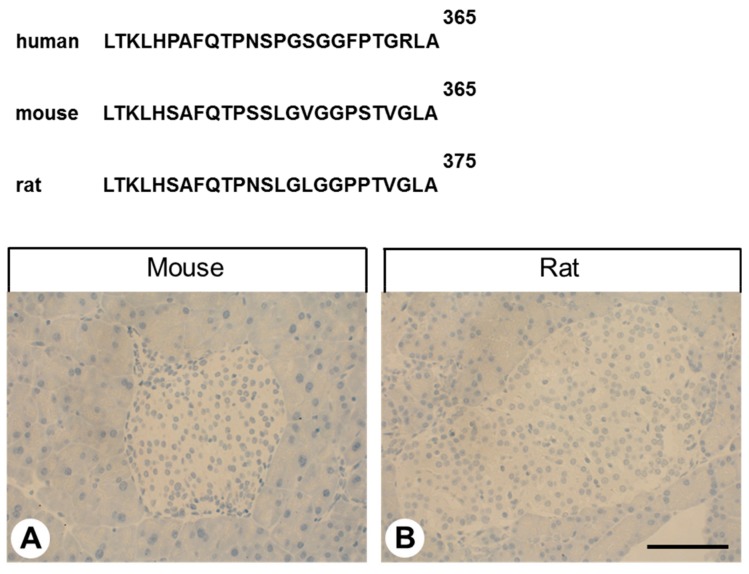
Carboxy-terminal sequence of human, mouse and rat GPR68 and negative GPR68-immunostaining in mouse and rat pancreatic islets. Immunohistochemical staining (red-brown colour), counterstaining with haematoxylin. Scale bar: 100 µm (**A**,**B**).

**Table 1 ijms-20-05261-t001:** Presence of GPR68 in different tumour entities.

Tumour Type (total number of cases)	GPR68 Positive Tumours (*n*)	Immunoreactive Score (IRS)			Tumours with GPR68 Positive Capillaries (*n*)
		mean	min	max	
Glioblastoma (10)	2	0.20	0	1	1
Pituitary adenoma (6)	3	1.50	0	3	1
- GH producing (4)	3	2.25	0	3	1
- ACTH producing (2)	0	0	0	0	0
Thyroid carcinoma (36)	10	0.67	0	4	1
- papillary (10)	3	0.40	0	2	0
- follicular (10)	0	0	0	0	0
- medullary (7)	6	1.70	0	3	0
- anaplastic (9)	1	0.89	0	4	1
Parathyroid adenoma (10)	0	0	0	0	0
Paraganglioma (10)	7	0.80	0	2	0
Lung cancer (20)	4	0.33	0	2	19
- Adenocarcinoma (10)	3	0.45	0	1.5	10
- Squamous cell carcinoma (10)	1	0.20	0	2	9
Gastric adenocarcinoma (19)	5	0.55	0	3	4
Colon carcinoma (10)	0	0	0	0	0
Gastrointestinal stromal tumour (10)	6	0.95	0	2	2
Hepatocellular carcinoma (10)	2	0.70	0	4	2
Cholangiocellular carcinoma (3)	0	0	0	0	0
Pancreatic adenocarcinoma (10)	6	1.80	0	3	0
Renal clear cell carcinoma (6)	0	0	0	0	2
Pheochromocytoma (6)	6	4.20	2	6	0
Neuroendocrine tumour (278)	49	1.47	0	12	27
Prostate adenocarcinoma (10)	0	0	0	0	3
Testicular cancer (10)	0	0	0	0	0
Breast carcinoma (10)	3	0.70	0	3	0
Endometrial cancer (10)	10	3.60	2	6	1
Cervical cancer (10)	4	1.50	0	6	3
- Adenocarcinoma (4)	4	3.75	3	6	1
- Squamous cell carcinoma (6)	0	0	0	0	2
Ovarian cancer (10)	3	0.50	0	2	1
Lymphoma (10)	0	0	0	0	3
Melanoma (5)	0	0	0	0	0
Sarcoma (14)	0	0	0	0	0
- Liposarcoma (4)	0	0	0	0	0
- Rhabdomyosarcoma (3)	0	0	0	0	0
- Leiomyosarcoma (4)	0	0	0	0	0
- Pleomorphic sarcoma (2)	0	0	0	0	0
- Osteosarcoma (1)	0	0	0	0	0

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
