# Peer review of "Comprehensive Assessment of GPR68 Expression in Normal and Neoplastic Human Tissues Using a Novel Rabbit Monoclonal Antibody"

_ijms, 2019, doi:10.3390/ijms20215261_

Round 1

Reviewer 1 Report

The authors generated and characterized a novel rabbit monoclonal anti-human GPR68 antibody. They demonstrated the antibody specificity by western blot using BON-1 cells treated with specific GPR68-siRNA and by immunocytochemical analysis on BON-1 exposed to different pH levels, showing corroboration with previous reports. Then, they used the anti-GPR68 antibody to examine the receptor expression in a broad range of normal human tissues and tumors by immunohistochemical analysis and demonstrated that GPR68 was most predominantly expressed in neuroendocrine tumours where it may be a positive prognostic factor and may act as a tumour suppressor. In conclusion, they claimed that the novel antibody will be a valuable tool for basic research and for identifying GPR68-expressing tumours during histopatological examinations. However, study is suffering from many shortcomings and the data are not completely supported by satisfying controls. Moreover, the data indicating GPR68 as a favourable prognostic factor and tumour suppressor should be supported by in vitro tumorigenic assays.

Major revisions:

Additional experiments with appropriate controls should be performed in order to validate the antibody specificity and the GPR68-positive prognostic behaviour:

the GPR68 subcellular localization should be performed in both HEK-293 and BON-1 over-expressing epitope-tagged-GPR68 with either novel rabbit monoclonal anti-GPR68 or epitope-tag antibodies BON-1 over-expressing endogenous GPR68 and some GPR68-positive tumours should be immunocytochemical/immunohistochemical analyzed with both novel anti-GPR68 and GPR68-commercial antibodies GPR68 gain-loss of function and tumorigenic in vitro assays (colony formation, invasion, migration, proliferation assays) should be employed to validate the prognostic features of GPR68

Minor revisions:

Quantitative PCR should be performed on BON-1 cells interfered with specific GPR68 siRNA in order to further validate the results obtained from western blot. Nuclei should be stained in the immunofluorescence analysis shown in figure 2. Panel A and B in figure 3 were not provided of the relative insets. The authors should mention the cell lines where GPR68 expression was devoid and they should describe them in material and methods section.

The paragraph 2.4.1 “patient characteristics” present in results section should be shift to material and methods section.

Reviewer 2 Report

In the present study, the authors have generated a rabbit monoclonal antibody that recognizes GPR68 protein.  GPR68 belongs to the family of proton-sensing G protein-coupled receptors involved in cellular pH sensing and cell adaptation to the changing pH environment upon cancer progression. The authors demonstrate that this antibody is suitable for Western blotting analysis, immunocytochemistry and immunofluorescent staining of normal and pathological tumor samples as well as of tumor cell lines. The antibody has been used for screening of GPR68 expression in a large variety of normal and cancerous human tissue samples. Furthermore, the results of pathological tissue staining have been correlated with clinical data. Although the study is comprehensive and involves a great amount of experimental work, some conclusions seem to be little substantiated.

The “Discussion” part as well as the “Results” contain the statements “Independent of tumour cells, capillaries, granulocytes, and macrophages within some tumours were strongly GPR68-positive.” This conclusion would need double immunofluorescent or immunohistochemical staining with antibodies, recognizing GPR68 and markers of endothelial cell/ granulocytes/macrophages. The same comment is true for the phrase “In the present study, GPR68 expression was also observed in bone marrow mononuclear cells, likely monocytes/macrophages and/or myeloid stem cells that can give rise to osteoclasts.” These data are not presented in the manuscript and needs further confirmation by double staining. Another phrase “GPR68 was also expressed in the smooth muscle layers of arteries and arterioles, and in capillaries….”should also be changed, otherwise the authors should provide convincing date including higher magnification clearly demonstrating GPR68 expression in different layers of blood vessels. The statement “In the current study, GPR68 expression was occasionally also observed in osteoblasts and osteocytes.”in the “Discussion” part is not substantiated by any presented data.

Scales on the figures will also contribute to a better understanding and presentation of the data. It seems that magnification in the subset of figures in Figure 5 and 6 is different.

The conclusion drawn at the end of the manuscript “Our results also illustrate the diverse and important roles of GPR68, such as those during hormone secretion, glucose homeostasis, regulation of bicarbonate and mucus secretion in the gastrointestinal tract, modulation of inflammatory processes, angiogenesis, and tumour growth.” is premature and can’t be drawn from the presented results. The obtained data allow to make an assumption about a possible connection and correlation, since no causal relationship has been detected.

Round 2

Reviewer 1 Report

No further comments